# Seasonal Variability of a Caryophyllane Chemotype Essential Oil of *Eugenia patrisii* Vahl Occurring in the Brazilian Amazon

**DOI:** 10.3390/molecules27082417

**Published:** 2022-04-08

**Authors:** Ellen de Nazaré S. da Cruz, Luana de Sousa Peixoto, Jamile S. da Costa, Rosa Helena V. Mourão, Walnice Maria O. do Nascimento, José Guilherme S. Maia, William N. Setzer, Joyce Kelly da Silva, Pablo Luis B. Figueiredo

**Affiliations:** 1Programa Institucional de Bolsas de Iniciação Científica, Universidade Federal do Pará, Belem 66075-900, Brazil; ellen.cruz@icen.ufpa.br; 2Laboratório de Química dos Produtos Naturais, Centro de Ciências Biológicas e da Saúde, Universidade do Estado do Pará, Belem 66087-662, Brazil; luana.peixoto@aluno.uepa.br (L.d.S.P.); jamile.costa@ics.ufpa.br (J.S.d.C.); 3Programa de Pós-Graduação em Ciências Farmacêuticas, Universidade Federal do Pará, Belem 66075-900, Brazil; gmaia@ufpa.br; 4Laboratório de Bioprospecção e Biologia Experimental, Universidade Federal do Oeste do Pará, Santarem 68035-110, Brazil; mouraorhv@yahoo.com.br; 5Laboratório de frutíferas, Embrapa Amazônia Oriental, Belem 66095-100, Brazil; walnice.nascimento@embrapa.br; 6Programa de Pós-Graduação em Química, Universidade Federal do Maranhão, Sao Luis 65080-805, Brazil; 7Aromatic Plant Research Center, 230 N 1200 E, Suite 100, Lehi, UT 84043, USA; setzerw@uah.edu (W.N.S.); joycekellys@ufpa.br (J.K.d.S.); 8Programa de Pós-Graduação em Biotecnologia, Universidade Federal do Pará, Belem 66075-900, Brazil; 9Departamento de Ciências Naturais, Centro de Ciências Sociais e Educação, Universidade do Estado do Pará, Belem 66087-662, Brazil

**Keywords:** Myrtaceae, essential oil composition, seasonal variation, chemometrics, sesquiterpenes

## Abstract

*Eugenia patrisii* Vahl is a native and non-endemic myrtaceous species of the Brazilian Amazon. Due to few botanical and phytochemical reports of this species, the objective of the present work was to evaluate the seasonal variability of their leaf essential oils, performed by GC and GC-MS and chemometric analysis. The results indicated that the variation in oil yields (0.7 ± 0.1%) could be correlated with climatic conditions and rainy (R) and dry seasons (D). (*E*)-caryophyllene (R = 17.1% ± 16.0, D = 20.2% ± 17.7) and caryophyllene oxide (R = 30.1% ± 18.4, D = 14.1% ± 19.3) are the major constituents and did not display significant differences between the two seasons. However, statistically, a potential correlation between the main constituents of *E. patrisii* essential oil and the climatic parameters is possible. It was observed that the higher temperature and insolation rates and the lower humidity rate, which are characteristics of the dry season, lead to an increase in the (*E*)-caryophyllene contents, while lower temperature and insolation and higher humidity, which occur in the rainy season, lead to an increase in the caryophyllene oxide content. The knowledge of variations in the *E. patrisii* essential oil composition could help choose the best plant chemical profile for medicinal purposes.

## 1. Introduction

The Myrtaceae comprises about 140 genera and 7000 species occurring mainly in the southern hemisphere, standing out in diverse habitats such as Australia, Southeast Asia, and tropical and subtropical Americas [1,2]. *Eugenia* genus comprises about 1000 species, one of the most representative of Myrtaceae, with occurrence in Central and South America and the African continent [3]. In addition to its high taxa, this genus has excellent economic and pharmacological potential [4].

*Eugenia patrisii* Vahl [syn. *Eugenia berlynensis* O. Berg., *E. inocarpa* DC., *E. parkeriana* DC., *E. teffensis* O. Berg., *E. vellozii* O. Berg., *Stenocalyx patrisii* (Vahl) O. Berg.] [5] is a native and non-endemic species of Brazil which may occur as a tree or shrub from 5 to 8 m tall, popularly known as ubaia, araçarana and fruta-de-jaboti [6,7,8]. It is considered a wild plant in the Brazilian Amazon, including the Guianas, Bolivia, and Peru. Their fruits are used to prepare refreshments, juices, and sweets. However, they still belong to the “wild fruits” category due to their lack of rational cultivation in the Amazon region [7]. Moreover, the infusion of their leaves, stems, and fruits is used practically by indigenous communities in the Amazon region to treat coughs and respiratory diseases [9].

The chemical composition of *E. patrisii* essential oils from the Amazon Region has been poorly investigated. However, the literature has indicated sesquiterpenes with cadinane and caryophyllane skeletons as the predominant classes in their essential oils [10]. Some studies have reported the cytotoxic potential of the essential oil of *E. patrisii*, rich in (*E*)-caryophyllene (32.0%) and bicyclogermacrene (10.0%), against melanoma, gastric, and colon cancer cells [11]. (*E*)-Caryophyllene and caryophyllene oxide are bicyclic sesquiterpene compounds with significant anesthetic, antiproliferative and cytotoxic effects evaluated in vitro and in vivo assays [12].

The main factors that can modify essential oil production and chemical composition are seasonality, circadian rhythm, stage of development, temperature, water availability, ultraviolet radiation, and others [13]. In this sense, in the last years, research has evaluated the influence of these factors on the *Eugenia* intra- and interspecific chemical variability. This knowledge is essential for essential oil standardization and their future economic and commercial applications [10].

Therefore, the objective of the present work was to evaluate the seasonal variability and the production of this caryophyllane chemotype of *E. patrisii* essential oil, with pharmacological potential as an anesthetic and anticancer agent, using chemometric tools.

## 2. Results and Discussion

### 2.1. Essential Oil Yield vs Environmental Condition

Climatic parameters, such as the monthly average of temperature, insolation, precipitation, and relative humidity, were monitored for one year (August 2020 to July 2021) to evaluate the seasonal influence on the yield and leaf essential oil composition of *E. patrisii*. Mean values of insolation ranged from 103.4 h (February) to 287.8 h (August), relative humidity from 75.5% (August) to 90.8 and 90.2% (February and March), average temperature from 26.0 °C (February) to 28.2 °C (August), and the average rainfall from 54.3 mm (August) to 614.1 mm (February). Based on the precipitation data, the rainy season occurred from January (328.3 mm) to May (451.2.4 mm), and the dry season from June (212.4 mm) to October (266.0 mm). In addition, November (502.1 mm) and December (213.6 mm) was the period of transition between these two seasons (Figure 1).

The climate of the Brazilian Amazon comprises only two well-defined seasons, the dry and rainy seasons, considering the stability of its hot and humid climate. The city of Belém, where the *E. patrisii* specimen was collected, is situated in the Amazon region of northern Brazil, which presents higher precipitation from December to April (rainy period), and lower precipitation from June to November (dry period). Complementarily, the months of June and July and December and January are considered transition months between these two seasonal periods [14,15].

In the seasonal study, the oil yield of *E. patrisii* ranged from 0.5% (Jun) to 1.0% (August), with an average of 0.7 ± 0.1% during the period (see Table 1). Statistically (Tukey test), there was no significant difference in the oil yields in the dry (0.7 ± 0.2%) and rainy (0.6 ± 0.1%) seasons. Additionally, a significant correlation (*p* > 0.05) was not observed between the essential oil content and the climatic parameters.

However, analyzing the Pearson correlation coefficient (r), the precipitation (r = −0.24), and insolation (r = 0.24) showed a weak negative and positive relationship with the essential oil yield, respectively. Moreover, temperature (r = 0.33) showed a weak positive value, while relative humidity (r = −0.42) displayed a weak negative relationship with the essential oil yield. Thus, the results indicate that the variation in oil yields of *E. patrisii* can be correlated with the climatic conditions and annual seasonality. It was impossible to distinguish the oil yields concerning the collection periods in the two seasons.

### 2.2. Oil Composition vs Seasonal Condition

GC-MS and GC-FID identified and quantified the constituents of the oils from *E. patrisii* leaves. Ninety-three volatile compounds were identified, representing an average of 90.3% of the total composition of the oils (Table 1). The sesquiterpene hydrocarbons (8.0–86.6%) and oxygenated sesquiterpenoids (4.0–78.7%) were the predominant constituents, followed by small amounts of other compounds (0–0.9%), monoterpene hydrocarbons (0–0.3%) and oxygenated monoterpenes (0–0.2%).

(*E*)-Caryophyllene (0–48.3%) and caryophyllene oxide (0–49.0%), caryophyllane-type sesquiterpenes, were the principal constituents of the *E. patrisii* oil during the seasonal period, followed by bicyclogermacrene (0–17.0%), spathulenol (0–11.4%), β-elemene (0.9–11.3%), and 9-*epi*-(*E*)-caryophyllene (0–7.3%). (*E*)-Caryophyllene was absent in February and March and showed a higher content in November (48.3%). Caryophyllene oxide was absent from August to November and showed a higher content in March (49.0%). Bicyclogermacrene, a sesquiterpene hydrocarbon with a germacrane skeleton, was absent from January to March and June to July, presenting its higher content in October (17.0%). Spathulenol, an aromadendrane-type oxygenated sesquiterpene, was absent in October and exhibited a higher content in March (11.4%). β-Elemene, a germacrane-type sesquiterpene, was present in all oils, with the higher content in January (11.3%). 9-*epi*-(*E*)-Caryophyllene, another sesquiterpene of caryophyllane-type, was absent in June and displayed a higher content in September (7.3%).

**Table 1 molecules-27-02417-t001:** Seasonal variability of the leaf essential oil of *Eugenia patrisii*: yields and oils constituents.

Oils Yields (%)			1.0	0.7	0.7	0.6	0.7	0.6	0.9	0.7	0.6	0.5	0.5	0.8
Constituents (%)	RI_C_	RI_L_	Aug	Sep	Oct	Nov	Dec	Jan	Fev	Mar	Apr	May	Jun	Jul
			Dry Season			Transition Months		Rainy Season					Dry Season	
Hexan-3-one	788	782 ^b^		0.1		0.3	0.1							
Hexan-2-one	792	786 ^b^		0.1	0.1	0.7	0.1						0.1	
Hexan-3-ol	796	795 ^b^		0.1		0.2								
*n*-Octane	799	800 ^a^					0.1	0.1			0.1			0.1
Hexanal	801	801 ^a^		0.1	0.1	0.9								
Styrene	892	891 ^b^		0.1	0.1	0.5	0.1							
Acetylacetone	922	919 ^a^		0.1	0.1	0.6								
α-Pinene	932	932 ^a^	0.1	0.1	0.1		0.1	0.1	0.1		0.1			0.1
β-Pinene	975	974 ^a^		0.1	0.1									
Limonene	1026	1024 ^a^		0.1									0.1	0.1
Linalool	1097	1095 ^a^	0.1	0.1	0.1				0.1					0.1
Nonanal	1101	1100 ^a^	0.1	0.1	0.1									
(3*Z*)-Hexenyl butanoate	1184	1184 ^a^	0.1	0.1	0.1		0.1							
Geranial	1269	1264 ^a^							0.1	0.1				
δ-Elemene	1335	1335 ^a^	1.6	2.7	1.1	0.4	0.8				0.3	0.3	0.1	0.1
α-Cubebene	1348	1345 ^a^	0.3	0.2	0.1		0.1				0.1	0.1	0.1	
Cyclosativene	1369	1369 ^a^									0.1		0.1	0.1
α-Ylangene	1373	1373 ^a^			0.1						0.1		0.1	
α-Copaene	1374	1374 ^a^	5.2	3.8	2.7	3.2	3.1	3.2	1.8	1.7	4.2	4.0	2.5	1.9
Isoledene	1375	1374 ^a^	0.2	0.2	0.1						0.1	0.1		
(3*Z*)-Hexenyl hexanoate	1380	1378 ^a^	0.2	0.1	0.1		0.2						0.1	0.1
β-Bourbonene	1387	1387 ^a^	0.7	0.3	0.2			0.4			0.2	0.2	0.2	0.2
β-Cubebene	1389	1387 ^a^	0.3	0.3	0.2									
β-Elemene	1390	1389 ^a^	6.9	1.7	2.7	3.7	3.9	11.3	2.5	0.9	2.5	2.3	5.3	7.1
β-Longipinene	1405	1400 ^a^		0.1										0.1
(*Z*)-Caryophyllene	1409	1408 ^a^			0.1							0.1		
α-Gurjunene	1410	1409 ^a^	1.4	0.7	0.4		0.4				0.4	0.4	0.1	
(*E*)-Caryophyllene	1419	1417 ^a^	31.6	30.1	37.3	48.3	34.0	22.2			31.5	31.6	1.9	0.1
β-Copaene	1430	1430 ^a^	0.3	0.3	0.2		0.1				0.2	0.2		
β-Gurjunene	1433	1431 ^a^	0.3	0.3	0.1		0.2				0.2	0.2	0.1	0.1
*trans*-α-Bergamotene	1434	1432 ^a^	1.3	1.3	1.0	0.4	0.8	0.5	0.3	0.4	1.2	1.2	0.6	0.4
Aromadendrene	1439	1439 ^a^	1.3	1.6	0.9		0.9	0.5	1.0	1.0	1.1	1.1	1.3	1.1
(*Z*)-β-Farnesene	1443	1440 ^a^			0.4			0.1					0.1	0.1
Myltayl-4(12)-ene	1444	1445 ^a^	0.6											
*trans*-Muurola-3,5-diene	1449	1451 ^a^		0.2	0.2		0.1							
α-Humulene	1452	1452 ^a^	4.4	4.8	4.5	3.8	3.7	3.0			3.7	3.7	0.6	0.1
(*E*)-β-Farnesene	1455	1454 ^a^		0.5	0.4		0.3				0.2	0.1		
9-*epi*-(*E*)-Caryophyllene	1464	1464 ^a^	5.4	7.3	6.3	7.2	6.6	3.1	2.5	2.7	7.1	6.8		2.2
*trans*-Cadina-1(6),4-diene	1473	1475 ^a^	0.2	0.5	0.3		0.2							
γ-Gurjunene	1475	1475 ^a^									0.1	0.1		
γ-Muurolene	1478	1478 ^a^	1.0	1.0	0.8		0.9	1.0	0.7	0.7	1.0	1.0	1.0	0.9
*trans*-β-Bergamotene	1482	1483 ^b^		0.3										
*ar*-Curcumene	1480	1479 ^a^											0.1	0.1
Germacrene D	1482	1484 ^a^	1.0	1.5	1.4		0.7				0.3	0.2		
β-Selinene	1489	1492 ^a^	0.6		0.4		0.4	1.1	0.1		0.4	0.4	1.1	1.0
*trans*-Muurola-4,(14),5-diene	1492	1493 ^a^	0.3	0.4	0.4		0.2							
*epi*-Cubebol	1494	1493 ^a^							0.4	0.4			0.7	0.4
α-Selinene	1494	1498 ^a^						1.2						
Bicyclogermacrene	1500	1500 ^a^	9.7	14.3	17.0	16.8	16.3				5.8	5.4		
α-Muurolene	1500	1500 ^a^		0.5	0.3		0.4	0.3	0.1	0.1	0.4	0.4	0.3	0.2
(*E*,*E*)-α-Farnesene	1506	1505 ^a^		1.2	1.3	0.6	1.0							
β-Bisabolene	1508	1505 ^a^	0.5					0.1			0.3	0.3	0.2	
α-Bulnesene	1508	1509 ^a^									0.1	0.1		
γ-Cadinene	1512	1513 ^a^	0.6	0.4	0.4									
β-Curcumene	1511	1514 ^a^		0.1	0.1									
Cubebol	1515	1514 ^a^						0.4	0.6	0.8	0.5	0.6	0.7	0.5
*trans*-Calamenene	1522	1521 ^a^						0.3					0.2	0.1
δ-Cadinene	1522	1522 ^a^	2.1	3.3	2.8	2.2	2.5				1.5	1.4		
*trans*-Cadina-1,4-diene	1532	1533 ^a^	0.1	0.2	0.2		0.1							
α-Cadinene	1537	1537 ^a^	0.1	0.1	0.1									
(*E*)-α-Bisabolene	1542	1540 ^a^	0.2	0.2	0.2		0.1					0.1		
Elemol	1548	1548 ^a^	0.1	0.1	0.1									
Germacrene B	1559	1559 ^a^	0.2	0.2	0.2		0.1	0.2	0.5	0.5	0.1	0.1		0.7
(*E*)-Nerolidol	1561	1561 ^a^		0.2	0.1		0.1							
Maaliol	1562	1566 ^a^					0.3						0.6	
Palustrol	1567	1567 ^a^	0.6	0.8	0.6		0.6		1.1	1.1	0.8	0.8	1.1	1.0
Caryophyllenyl alcohol	1573	1570 ^a^											0.2	
Spathulenol	1577	1577 ^a^	3.4	2.3		2.5	3.7	4.5	8.2	11.4	5.6	5.5	9.0	5.2
Caryophyllene oxide	1583	1582 ^a^					5.8	29.2	48.3	49.0	11.1	12.7	32.9	37.4
Globulol	1586	1590 ^a^	5.3	4.3	3.8									
Viridiflorol	1592	1592 ^a^	1.4	1.8	1.4	1.0	1.6	1.3	3.3	3.6	1.7	2.1	2.5	2.3
Cubeban-11-ol	1594	1595 ^a^	0.6	0.9	0.6		0.6	0.5	0.9	0.9	0.7	0.8	0.8	0.7
Ledol	1602	1602 ^a^	1.5	1.9	1.5		1.9	1.0	2.5	3.1	1.7	1.9	2.4	2.3
Humulene epoxide II	1608	1608 ^a^	0.3	0.1	0.1		0.1	2.0	3.4	3.3	0.9	0.9	3.5	3.5
1,10-*epi*-Cubenol	1616	1618 ^a^			0.1						0.1			
1-*epi*-Cubenol	1626	1627 ^a^	0.6	0.7	0.6		0.6	0.4	0.3	0.5	0.6	0.6	0.6	0.4
Muurola-4,10(14)-dien-1β-ol	1635	1630 ^a^		0.3	0.2			1.3						
*allo*-Aromadendrene epoxide	1636	1639 ^a^								0.3			0.3	0.3
Caryophylla-4(12),8(13)-dien-5β-ol	1638	1639 ^a^											0.4	0.5
*epi*-α-Muurolol	1640	1640 ^a^	1.1	1.2	1.0		1.0	0.9	0.6	0.7	1.2	1.3	1.0	0.6
α-Muurolol	1644	1644 ^a^	0.5	0.4	0.3		0.4	0.3	0.3	0.4	0.4	0.5	0.5	0.4
β-Eudesmol	1649	1649 ^a^	0.1								0.1	0.1		
α-Cadinol	1652	1652 ^a^	1.1	0.9	0.8	0.5	1.2			1.0	1.1	1.4		
Selin-11-en-4α-ol	1657	1658 ^a^			0.1				1.5				2.5	2.4
*cis*-Calamenen-10-ol	1658	1660 ^a^	0.1								0.2	0.2		
Intermedeol	1660	1665 ^a^							0.4					
14-Hydroxy-9-*epi*-(*E*)-caryophyllene	1664	1668 ^a^							2.9	1.6			2.7	
*trans*-Calamenen-10-ol	1667	1668 ^a^									0.1	0.1	0.2	0.1
Mustakone	1676	1676 ^a^							0.7	0.6	0.1	0.1	0.6	1.5
*epi*-α-Bisabolol	1681	1683 ^a^	0.1	0.1	0.1								0.1	
α-Bisabolol	1687	1685 ^a^	0.2	0.2	0.2		0.1				0.3	0.3		
14-Hydroxy-δ-cadinene	1807	1803 ^a^											0.4	
(2*Z*,6*E*)-Farnesyl acetate	1819	1821		0.1							0.1	0.1		
Monoterpene hydrocarbons			0.1	0.3	0.2		0.1	0.1	0.1		0.1		0.1	0.2
Oxygenated monoterpenoids			0.1	0.1	0.1				0.2	0.1				0.1
Sesquiterpene hydrocarbons			78.4	80.6	84.9	86.6	77.9	48.5	9.5	8.0	63.2	61.9	16.0	15.8
Oxygenated sesquiterpenoids			17.0	16.3	11.6	4.0	18.0	41.8	75.4	78.7	27.3	30.0	63.7	60.3
Others			0.4	0.9	0.7	3.2	0.7	0.1			0.1		0.2	0.2
Total (%)			96.0	98.2	97.5	93.8	96.7	90.5	85.2	86.8	90.7	91.9	80.0	76.6

RI_C_ = Calculated Retention Index (Rtx-5ms column); RI_L_ = Literature Retention Index; ^a^ = Adams, 2007 [16]; ^b^ = Mondello, 2011 [17]; Main constituents in bold, *n* = 2 (standard deviation was less than 2.0).

Analyzing the average content and standard deviation of the compound classes present in *Eugenia patrisii* essential oil on rainy (R) and dry (D) seasons (Figure 2), the sesquiterpene hydrocarbons (R = 38.1% ± 27.44, D = 54.9% ± 35.7) and oxygenated sesquiterpenoids (R = 50.5% ± 24.8, D = 33.6% ± 26.0), did not present statistically significant differences (Tukey test, *p* > 0.05). Likewise, (*E*)-caryophyllene (R = 17.1% ± 16.0, D = 20.2% ± 17.7), caryophyllene oxide (R = 30.1% ± 18.4, D = 14.1% ± 19.3), 9-*epi*-(*E*)-caryophyllene (R = 4.4% ± 2.3, D= 4.2% ± 3.0), spathulenol (R = 7.0% ± 2.8, D = 4.0% ± 4.0), bicyclogermacrene (R = 2.2% ± 3.1, D = 8.2% ± 7.2), and β-Elemene (R = 3.9% ± 4.2, D = 4.7% ± 2.4) did not display significant differences between the two seasons (Figure 2).

### 2.3. Oil Composition vs. Environmental Condition

The climate variables were correlated by the Pearson correlation coefficient (r) with the primary constituents of *E. patrisii* essential oil to verify their environmental conditions relationships (Table 2).

The parameters temperature, humidity and insolation correlated with the primary constituents of *E patrisii* essential oil. (*E*)-Caryophyllene displayed a moderate positive correlation with temperature (r = 0.65), and weak negative and positive correlations with humidity (r = −0.49) and insolation (r = 0.48). Additionally, caryophyllene oxide showed significant correlation with temperature (r = −0.85), humidity (r = 0.73) and insolation (r = −0.74). The seasonal rhythm of (*E*)-caryophyllene and caryophyllene oxide are shown in Figure 3. Two other sesquiterpenes displayed significant correlations: bicyclogermacrene with temperature (r = 0.72), and spathulenol with temperature (r = −0.76) and insolation (r = −0.71). Additionally, bicyclogermacrene (r = −0.59) and spathulenol (r = 0.61) showed moderate correlation with humidity (Table 2).

Concerning the (*E*)-caryophyllene and caryophyllene oxide contents, there was a strong and significant negative correlation between these two constituents (r = −0.93). Moreover, (*E*)-caryophyllene showed a strong and positive correlation with the sesquiterpene hydrocarbons 9-*epi*-(*E*)-caryophyllene (r = 0.89) and bicyclogermacrene (r = 0.85). Additionally, (*E*)-caryophyllene showed a strong and negative correlation with the oxygenated sesquiterpene spathulenol (r = −0.80). With respect to caryophyllene oxide, there was a significant and strong negative correlation with the sesquiterpene hydrocarbons 9-*epi*-(*E*)-caryophyllene (r = −0.80) and bicyclogermacrene (r = −0.85) and a strong positive correlation with spathulenol (r = −0.85).

These findings may be explained by the biosynthetic pathway of these constituents (Figure 4). Indeed, (*E*)-caryophyllene, 9-*epi*-(*E*)-caryophyllene, and caryophyllene oxide belong to the same pathway through the caryophyllyl cation. Caryophyllene oxide is a metabolic product of the (*E*)-caryophyllene oxidation, meaning that the increase in the caryophyllene oxide content implies decreasing (*E*)-caryophyllene content in the *E. patrisii* oils samples (Figure 2). Moreover, the caryophyllyl cation, the precursor of the caryophyllane-type sesquiterpenes, is different from the germacryl cation, which is the precursor of bicyclogermacrene and spathulenol.

(*E*)-Caryophyllene and caryophyllene oxide have a robust wooden odor, used as cosmetic and food additives. These two natural compounds are approved as flavorings by the Food and Drug Administration (FDA) and European Food Safety Authority (EFSA) and used in a mixture, how they often occur in plants. In medical practice, the application of the mixture of (*E*)-caryophyllene and caryophyllene oxide in combination with the classical anticancer drugs could bring significant benefits, potentializing the efficacy of the chemotherapeutics, eliciting the supplementary antineoplastic effect, and reducing the refractory cancer pain, at the same time [12]. It has been seen that among (*E*)-caryophyllene and caryophyllene oxide, this letter has more substantial anticancer properties, explained by its chemical structure. The caryophyllene oxide structure contains the functional groups methylene and exocyclic epoxide. Therefore, it can bind covalently to proteins and DNA bases by the sulfhydryl and amino groups. So, caryophyllene oxide has revealed high potential as a signaling modulator for tumor cancer cells [12,18].

The essential oils of Brazilian Myrtaceae have shown notable cytotoxic activity against the lung, colon, stomach, and melanoma cells, with a real prospect for subsequent phytotherapeutic development [11]. In seasonal studies, some Myrtaceae species have shown correlations of some essential oil constituents with environmental conditions. The sesquiterpene hydrocarbon δ-cadinene, one of the main constituents of the essential oil from *Myrcia sylvatica* (G. Mey) DC., with occurrence in the Amazon, displayed a significant and moderate correlation with the temperature (r = −0.6) and relative humidity (r = 0.7). Additionally, a moderate correlation between these two parameters was observed with oxygenated sesquiterpenoids and monoterpenes hydrocarbons [15]. In addition, the essential oil from *Calycolpus goetheanus* (Mart. ex DC.) O. Berg., from Marajo Island, Pará state, Brazilian Amazon, showed a strong positive correlation between temperature and monoterpene hydrocarbons (r = 0.75), a significant negative correlation of the sesquiterpene hydrocarbons with solar radiation (r = −0.72), and a positive correlation between the oxygenated sesquiterpenoids and solar radiation (r = 0.75) [19].

However, the high content of curzerene, the main constituent of the essential oil from *Eugenia uniflora* L., did not present a statistically significant difference during the dry (42.7% ± 6.1) and rainy (40.8 ± 5.9%) seasons. Furthermore, the climatic variables, relative humidity (r = −0.07), solar radiation (r = −0.19), and precipitation (r = −0.03), showed no significant correlation with curzerene content [14]. The essential oil of *Psidium myrtoides* O. Berg. presented the highest yields in the dry season (0.96–1.02%), decreasing in the rainy season (0.36–0.48%). A negative correlation between the oil yield and rainfall was noted. The predominant constituents in these two seasons were 1,8-cineole (29.5–48.1%), α-eudesmol (11.7–20.0%), α-pinene (5.0–12.8%), elemol (3.3–6.7%), and γ-eudesmol (2.5–5.8%) [20]. In addition, the rainy season was considered the ideal period for the oil extraction of *Psidium salutare* (Kunth). O. Berg., which showed higher oil yields. The main oil constituents were *p*-cymene (5.1–17.8%), terpinolene (6.9–17.0%), γ-terpinene (10.3–17.1%), *epi*-α-cadinol (10.4–12.8%), linalool (4.7–7.3%), and δ-cadinene (3.8–5.3%), showing absence of a statistical correlation with the environmental parameters [21].

Moreover, *Psidium myrtoides* oil presented the highest yields in the dry season (0.96–1.02%), decreasing in the rainy seasons (0.36–0.48%). A negative correlation between the oil yield and rainfall was noted. The most predominant constituents in these two seasons were 1,8-cineole (29.5–48.1%), α-eudesmol (11.7–20.0%), α-pinene (5.0–12.8%), elemol (3.3–6.7%), and γ-eudesmol (2.5–5.8%) [20]. Thus, the rainy season was considered the ideal period for the oil extraction of *Psidium salutare*, which showed the highest result. The main oil constituents were *p*-cymene (5.1–17.8%), terpinolene (6.9–17.0%), γ-terpinene (10.3–17.1%), *epi*-cadinol (10.4–12.8%), linalool (4.7–7.3%), and δ-cadinene (3.8–5.3%), showing not a statistical correlation with the environmental parameters [21].

These findings suggest a statistical correlation between the oils’ main constituents and the climatic parameters in some Myrtaceae species and the variation in the composition of the oils, which may be related to climatic factors. In general, it was observed that the higher temperature and insolation rates and the lower humidity rate, which are characteristics of the dry season, lead to an increase in the content of (*E*)-caryophyllene, while the lower temperature and insolation rates, and higher humidity rate leads to an increase in the caryophyllene oxide.

Application of a hierarchical clustering analysis (HCA) provided the dendrogram presented in Figure 5, which shows that the compositions of the analyzed oils were included in two different groups and presented a similarity of 0%. The hierarchical clustering analysis (HCA) and principal component analysis (PCA) were plotted with the constituents of oils which displayed values above 5%.

Group I, with 25.3% similarity between the samples, comprises the leaf oils collected in January, February, March, June, and July, whose main constituent was caryophyllene oxide. Group II includes oil samples from April, May, August, September, October, November, and December, which presented (*E*)-caryophyllene as the main component, showing 46.1% similarity to each other. The HCA plot shows that the two groups were formed differently during the collection period.

In the principal component analysis (PCA, Figure 6), the main components (PC1, PC2 and PC3) accounted for 89.8% of the total data variability. PC1 explained 66.3% and displayed positive correlations with the variables δ-elemene (r = 0.31), α-copaene (r = 0.31), (*E*)-caryophyllene (r = 0.40), 9-*epi*-(*E*)-caryophyllene (r = 0.39) and bicyclogermacrene (r = 0.39). PC2 explained 15.5% and displayed positive correlations with the variables β-elemene (r = 0.87), α-copaene (r = 0.30), (*E*)-caryophyllene (r = 0.06). The third component PC3, explained 8.0% and displayed positive correlations with the variables δ-elemene (r = 0.52), α-copaene (r = 0.63), and spathulenol (r = 0.37). Additionally, caryophyllene oxide showed negative correlation in PC1 (r = −0.42) and PC2 (r = −0.07).

PCA and HCA analyses showed no separation between the oil samples of *E. patrisii* during the dry and rainy periods. However, correlations between the oils’ constituents and the environmental parameters were observed. Some studies in Myrtaceae have shown that variation in the chemical composition of essential oils of *Eugenia*, *Syzygium*, and *Psidium* species could be affected by seasonality [10,21,22]. On the other hand, these variations sometimes are not distinguished by PCA and HCA analyses. For example, the seasonal study of *Eugenia uniflora* showed no separation between the oil samples from the dry and rainy periods in the PCA and HCA analyses [14].

## 3. Material and Methods

### 3.1. Plant Material and Climatic Data

The leaves of *E. patrisii* were collected from a specimen existing in Belém city, Pará state, Brazil (coordinates: 1°26′14.20″ S/48°26′30.20″ W). For the seasonal study, the mature leaves (1 kg) were sampled on day 10 of each month, at 8 am, from August 2020 to July 2021. Plant identification was performed by comparison with an authentic specimen of *Eugenia patrisi* Vahl (MG227357), and a plant sample was incorporated into the Herbarium “João Murça Pires”, at Museu Paraense Emílio Goeldi, Belém city, State of Pará, Brazil. At the same time, the climatic parameters (insolation, relative air humidity, and rainfall precipitation) of the mentioned area were obtained each month from the website of the Instituto Nacional de Meteorologia (INMET, http://www.inmet.gov.br/portal/, accessed on 31 March 2022, of the Brazilian Government [23]. Meteorological data were recorded through the automatic station A-201, located in Belém city, Pará state, Brazil, equipped with a Vaisala system, model MAWS 301 (Vaisala Corporation, Helsinki, Finland).

### 3.2. Extraction and Oil Composition

The air-dried leaves (150 g) were ground and subjected to hydrodistillation in duplicate using a Clevenger-type apparatus (3 h). The oils were dried over anhydrous sodium sulfate, and the dry plant weights were used to calculate their yields. The moisture content of the plant samples was calculated in an infrared moisture balance for water loss measurement. Analysis of oil yield was done in triplicate. The essential oil was dissolved in *n*-hexane (1500 µg/mL, 3:500, *v*/*v*), and analyzed simultaneously in these two systems: gas chromatography–flame ionization detector (GC-FID, Shimadzu Corporation, Tokyo, Japan) and gas chromatography-mass spectrometry (GC/MS, Shimadzu Corporation, Tokyo, Japan). The oil analyses were performed in a GCMS-QP2010 Ultra system (Shimadzu Corporation, Tokyo, Japan), equipped with an AOC-20i auto-injector and the GCMS-Solution software containing the Adams and FFNSC-2 libraries [16,17]. An Rxi-5ms (30 m; 0.25 mm; 0.25 µm film thickness) silica capillary column (Restek Corporation, Bellefonte, PA, USA) was used. The conditions of analysis were as follows: injector temperature: 250 °C; oven temperature programming: 60–240 °C (3 °C/min); helium as the carrier gas, adjusted to a linear velocity of 36.5 cm/s (1.0 mL/min); split mode injection for 1.0 µL of essential oil solution (6 µg of essential oil injected); split ratio 1:20 (300 ng to column for analysis); ionization by electronic impact at 70 eV; ionization source and transfer line temperatures of 200 °C and 250 °C, respectively. The mass spectra were obtained by automatic scanning every 0.3 s, with mass fragments in the range of 35–400 m/z. The retention index was calculated for all volatile components using a homologous series of C_8_-C_40_ *n*-alkanes (Sigma-Aldrich, Milwaukee, WI, USA), according to the linear equation of Van Den Dool and Kratz [24]. Individual components were identified by comparing their retention indices and mass spectra (molecular mass and fragmentation pattern) with those in the GCMS-Solution system libraries [16,17]. The quantitative data regarding the volatile constituents were obtained using a GC 2010 Series, operated under similar conditions to the GC-MS system. The relative amounts of individual components were calculated by peak-area normalization using the flame ionization detector (GC-FID). GC-FID and GC-MS analyses were performed in duplicate.

### 3.3. Statistical Analysis

Statistical significance was assessed by a Tukey test (*p* < 0.05), and the Pearson correlation coefficients (r) were calculated to determine the relationship among the parameters analyzed (insolation, relative air humidity, and rainfall precipitation), using the software GraphPad Prism, version 5.0. The principal component analysis (PCA) was applied to verify the interrelation in the oils’ components (>5%) (OriginPro trial version, OriginLab Corporation, Northampton, MA, USA). The hierarchical grouping analysis (HCA), considering the Euclidean distance and complete linkage, was used to verify the similarity of the oil samples based on the distribution of the constituents selected in the previous PCA analysis.

## 4. Conclusions

The results showed that climatic variables are correlated with the yield and chemical constituent concentrations of *E. patrisii* essential oil. Environmental factors also had a strong influence on the composition of the oil.

The ecological significance of the essential oil variations is not yet obvious. It is not clear if the responses are due directly to abiotic factors such as sunlight, temperature, humidity, or rainfall, or to secondary pressures related to seasonality such as herbivory, microbial infections, or phenology. Further research is needed at this point.

The high (*E*)-caryophyllene and caryophyllene oxide content in this *E. patrisii* specimen suggests its potential as a renewable source of these biologically active compounds, as an anticancer agent and anesthetic. This study contributes better to the knowledge of *E. patrisii* essential oil, already used for its medicinal potential. The knowledge of variability in the chemical composition of the *E. patrisii* essential oil can help choose the appropriate chemical profile for medicinal purposes.

## Figures and Tables

**Figure 1 molecules-27-02417-f001:**
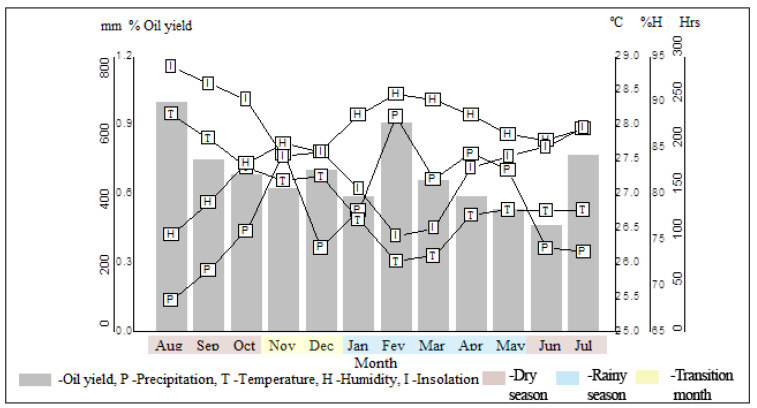
Environmental conditions vs. *Eugenia patrisii* essential oil yield in the seasonal study.

**Figure 2 molecules-27-02417-f002:**
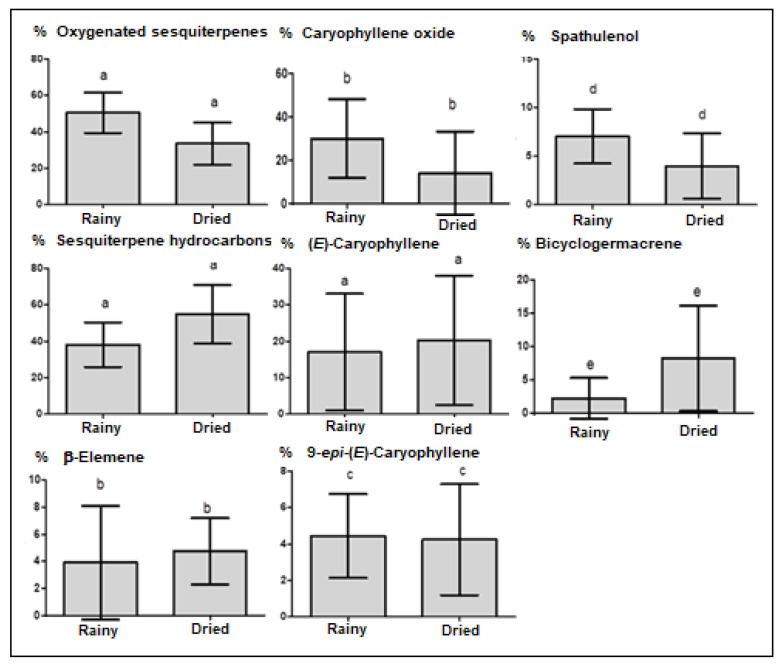
Compound classes and primary constituents of the *E. patrisii* essential oils on rainy and dried seasons: media and standard deviation. Values with the same letters in the bars do not differ statistically in the Tukey test (*p* > 0.05).

**Figure 3 molecules-27-02417-f003:**
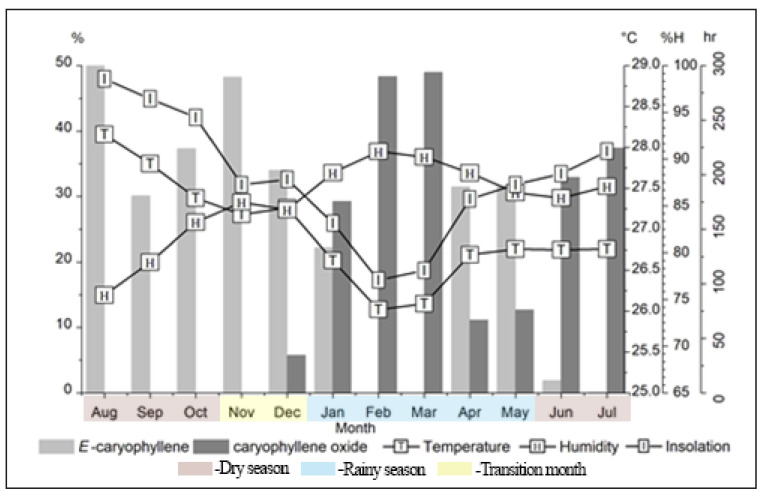
The seasonal rhythm of (*E*)-caryophyllene and caryophyllene oxide, the main constituents of *E. patrisii* essential oil.

**Figure 4 molecules-27-02417-f004:**
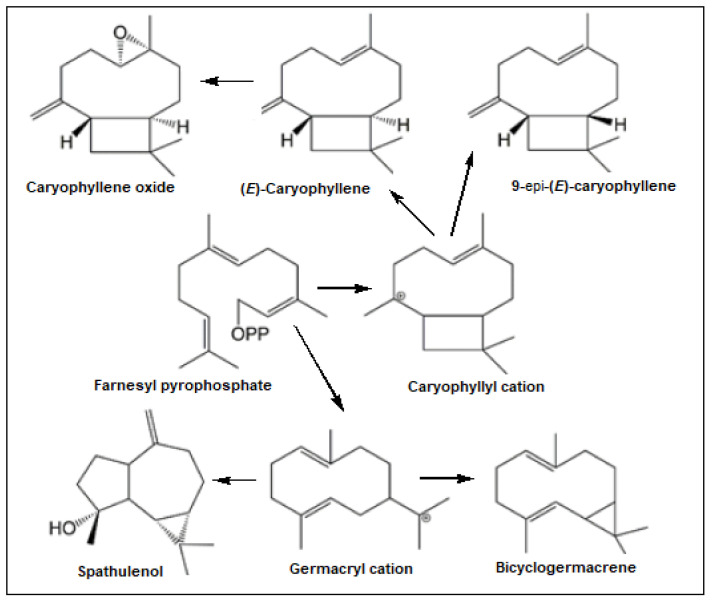
Biosynthetic pathway of the main constituents from *E. patrisii* essential oil.

**Figure 5 molecules-27-02417-f005:**
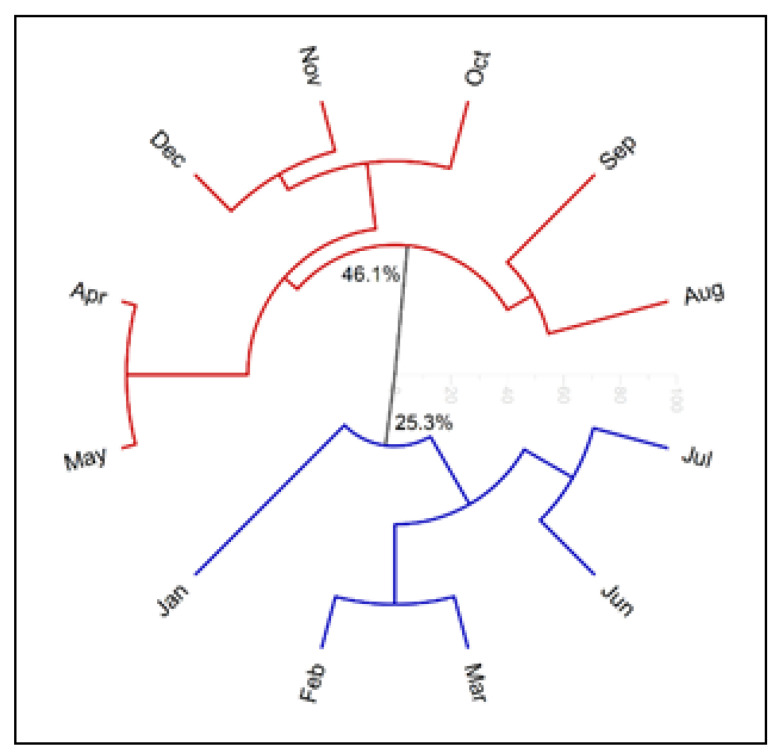
Hierarchical clustering analysis (HCA) based on the main constituents (above 5%) of *E. patrisii* essential oils.

**Figure 6 molecules-27-02417-f006:**
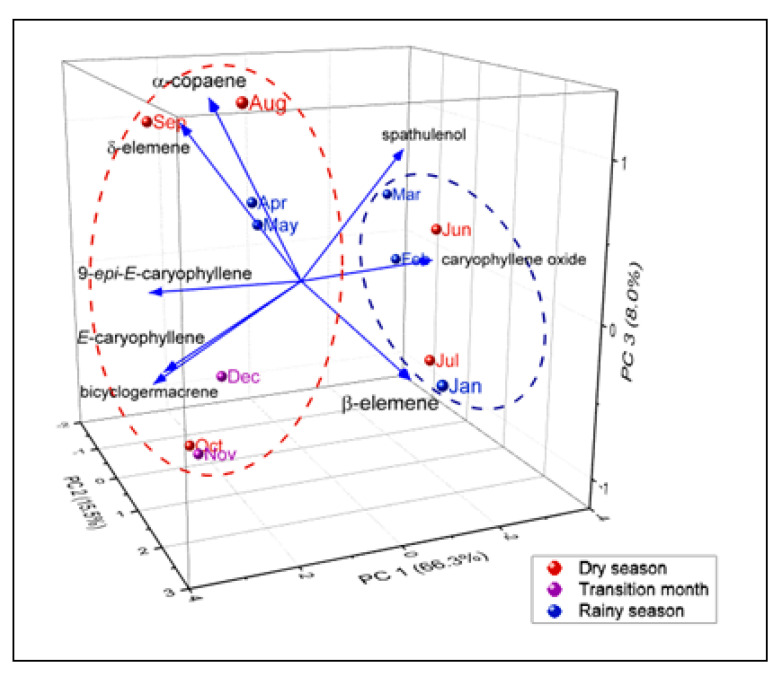
Principal component analysis (PCA) based on the primary constituents (above 5%) of *E. patrisii* essential oils.

**Table 2 molecules-27-02417-t002:** Correlation of *E. patrisii* essential oil primary constituents with the collection site climatic parameters.

Constituents	Precipitation	Temperature	Humidity	Insolation
(*E*)-Caryophyllene	−0.07	0.65 *	−0.49	0.48
Caryophyllene oxide	0.36	−0.85 *	0.73 *	−0.74 *
9-*epi*-(*E*)-Caryophyllene	0.03	0.54	0.41	0.40
Bicyclogermacrene	−0.25	0.72 *	−0.59 *	0.57 *
Spathulenol	0.36	−0.76 *	0.61 *	−0.71 *
β-Elemene	−0.37	0.12	−0.08	0.13

* Significant at *p* < 0.05.

## Data Availability

Not applicable.

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
