# Peer review of "Seasonal Variability of a Caryophyllane Chemotype Essential Oil of Eugenia patrisii Vahl Occurring in the Brazilian Amazon"

_molecules, 2022, doi:10.3390/molecules27082417_

Round 1

Reviewer 1 Report

In the paper Seasonal Variability of a Caryophyllane Chemotype Essential Oil of Eugenia patrisii Vahl Occurring in the Brazilian Amazon authors report a study to evaluate the seasonal variability of their leaf essential oils, performed by GC and GC-MS and chemometric analysis. Report is interesting, good write. 
I suggest minor revisions.

Specific comments
Please add error bras in Figure 1. Environmental conditions vs. Eugenia patrisii essential oil yield in the seasonal study.
and in the Figure 3. The seasonal rhythm of (E)-caryophyllene and caryophyllene oxide, the main constituents 162
of E. patrisii essential oil.

Table 1
Please add other recent references to calculate RIC. 
In this context add for example 
Flavouring extra-virgin olive oil with aromatic and medicinal plants essential oils stabilizes oleic acid composition during photo-oxidative stress. Agriculture, 11(3), 266.
RIC = Calculated Retention Index (Rxi-5ms column); RIL = Literature Retention Index; a = Adams, 2007 [16]; b = Mondello, 2011 [17]; Main constituents in bold.

page 12 line 303 
The conditions of analysis were as follows. Injector 303
temperature: 250 °C
Please, for fiture work, consider to use a PTV injection system. 
In detail, several analytes can be degradate at this temperature of injection. 

Author Response

molecules-1675216

Responses to Reviewer #1

In the paper Seasonal Variability of a Caryophyllane Chemotype Essential Oil of Eugenia patrisii Vahl Occurring in the Brazilian Amazon authors report a study to evaluate the seasonal variability of their leaf essential oils, performed by GC and GC-MS and chemometric analysis. Report is interesting, good write.

I suggest minor revisions.

Response: We are very grateful for the comments of the reviewer 1.

Specific comments

Please add error bras in Figure 1. Environmental conditions vs. Eugenia patrisii essential oil yield in the seasonal study.

Response: By inserting the bar with the standard deviations of oil yields (Figure 1) and E-caryophyllene and caryophyllene oxide contents (Figure 2), both figures were very opulent and difficult to follow. Thus, the standard deviation data were less than 2.0%.  However, we improved the design of these figures following reviewer 2's suggestion.

and in the Figure 3. The seasonal rhythm of (E)-caryophyllene and caryophyllene oxide, the main constituents 162 of E. patrisii essential oil.

Response: See above

Table 1

Please add other recent references to calculate RIC. In this context add for example Flavouring extra-virgin olive oil with aromatic and medicinal plants essential oils stabilizes oleic acid composition during photo-oxidative stress. Agriculture, 11(3), 266.

Response: Dear Reviewer, the van der Doon and Kratz equation is the base reference to calculate the retention index. Moreover, in the article suggested by you, the authors used a DBWAX capillary column coated with polyethylene glycol (A polar stationary phase). In our work, we used a Rtx-5ms column coated with 5% diphenyl/95% dimethyl polysiloxane phase (a low polarity column). So, we could not replace this reference.

RIC = Calculated Retention Index (Rxi-5ms column); RIL = Literature Retention Index; a = Adams, 2007 [16]; b = Mondello, 2011 [17]; Main constituents in bold.

Response: see above

page 12 line 303 The conditions of analysis were as follows. Injector temperature: 250 °C

Response: see above

Please, for fiture work, consider to use a PTV injection system.  In detail, several analytes can be degradate at this temperature of injection.

Response: Thanks, we will consider to use a PTV injection system in our future works

Prof. Dr. Pablo Luis. B. Figueiredo

UEPA, Belém, Brazil

Reviewer 2 Report

This paper investigated the chemical composition of Eugenia patrisii and found that the variation in oil yields was correlated with climatic conditions; moreover, the authors proposed that there is a potential correlation between the main constituents such as (E)-caryophyllene and caryophyllene oxide and the climatic parameters. These findings are important in terms of revealing the relationship between the characteristics of plant derived essential oil and the environment. This paper can be accepted for publication, however some revisions are necessary to improve the quality of the manuscript:

  1. as the authors explained, 12 months are divided into the rainy period, the dry period and the transition months. I suggest the authors to clearly classify these different periods in table 1 and figure 1 so that readers are able to interpretate the data easily, for example, the rainy period months are bolded, or simply adding a line indicating which months are the dry period, etc.
  2. more details are needed about collection of the plant materials. It seems to me that all leaves were collected from the same plant; I suggest that more plant specimens to be collected from different sites to compare the results, which would be more convincing, at least in the future work. And how about the weight? Each time how many grams of leaves were harvested? And if possible, write about from which part of the plant the leaves were collected (upper? Bottom?) Were there any replicates (in the text it said that GC/MS was repeated twice; using the same leaf oil?)?
  3. I don’t understand why the authors did not extract essential oils from fresh leaves. Is there any particular reason why the leaves were air dried before extraction? The drying process might influence the results.
  4. “E. patrisii” needs to be italicized in the manuscript, for example in L95.
  5. if possible, please discuss the ecological significance of the variation of the essential oil in different seasons.

Author Response

molecules-1675216

Responses to Reviewer #2

This paper investigated the chemical composition of Eugenia patrisii and found that the variation in oil yields was correlated with climatic conditions; moreover, the authors proposed that there is a potential correlation between the main constituents such as (E)-caryophyllene and caryophyllene oxide and the climatic parameters. These findings are important in terms of revealing the relationship between the characteristics of plant derived essential oil and the environment. This paper can be accepted for publication, however some revisions are necessary to improve the quality of the manuscript:

Response: We are very grateful for the comments of the reviewer 2.

as the authors explained, 12 months are divided into the rainy period, the dry period and the transition months. I suggest the authors to clearly classify these different periods in table 1 and figure 1 so that readers are able to interpretate the data easily, for example, the rainy period months are bolded, or simply adding a line indicating which months are the dry period, etc.

Response: We remade Figures 1 and 3 and Table 1 to address this suggestion. Thanks for your attention

more details are needed about collection of the plant materials. It seems to me that all leaves were collected from the same plant; I suggest that more plant specimens to be collected from different sites to compare the results, which would be more convincing, at least in the future work. And how about the weight? Each time how many grams of leaves were harvested? And if possible, write about from which part of the plant the leaves were collected (upper? Bottom?) Were there any replicates (in the text it said that GC/MS was repeated twice; using the same leaf oil?)?

Response: Corrected. We collected about 1 kg of fresh mature leaves. After drying, about 150 g of dried leaves were hydrodistilled in duplicate.

I don’t understand why the authors did not extract essential oils from fresh leaves. Is there any particular reason why the leaves were air dried before extraction? The drying process might influence the results.

Response: Dear reviewer, the botanical material was collected far from our laboratory, so it would be logically impossible to extract the fresh material. Thus, the covid 19 pandemic limited our access in staggered days, which made the extraction of fresh material unfeasible. However, the material was dried following the same protocol in the 12 months of study, which removes the interference of drying in our results.

“E. patrisii” needs to be italicized in the manuscript, for example in L95.

Response: Corrected

if possible, please discuss the ecological significance of the variation of the essential oil in different seasons.

Prof. Dr. Pablo Luis. B. Figueiredo

UEPA, Belém, Brazil
